# HYBRID REINFORCEMENT:
# WHEN REWARD IS SPARSE, BETTER TO BE DENSE

**Leitian Tao**[1,2]*, **Ilia Kulikov**[1], **Swarnadeep Saha**[1], **Tianlu Wang**[1], **Jing Xu**[1], **Sharon Li**[2],
**Jason Weston**[1], **Ping Yu**[1]

[1]FAIR at Meta      [2]University of Wisconsin–Madison

## ABSTRACT

Post-training for reasoning in large language models has increasingly relied on verifiable rewards: deterministic checkers that provide 0–1 correctness signals. While reliable, such binary feedback is brittle—many tasks admit partially correct or alternative answers that verifiers under-credit, and the resulting all-or-nothing supervision limits learning. Reward models offer richer, continuous feedback, which can serve as a complementary supervisory signal to verifiers. We introduce HERO (**H**ybrid **E**nsemble **R**eward **O**ptimization), a reinforcement learning framework that integrates sparse verifier signals with dense reward model scores in a structured way. HERO employs stratified normalization to bound reward-model scores within verifier-defined groups, preserving correctness while refining quality distinctions, and variance-aware weighting to emphasize challenging prompts where dense signals matter most. Across diverse mathematical reasoning benchmarks, HERO consistently outperforms reward model-only and verifier-only baselines, with strong gains on both verifiable and hard-to-verify tasks. Our results show that hybrid reward design retains the stability of verifiers while leveraging the nuance of reward models to advance reasoning.

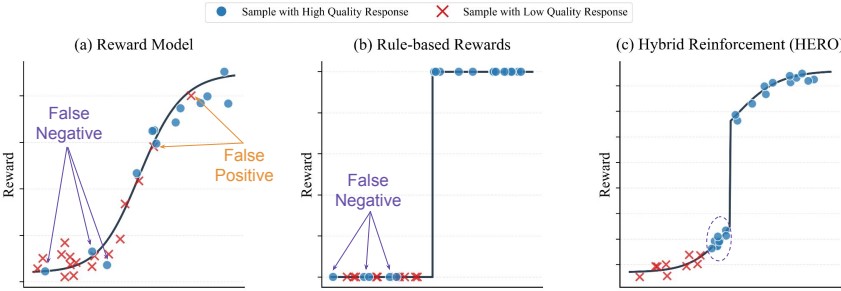

Figure 1: **Comparison of reward signals from different supervision sources.** Reward Models (a) provide smooth but sometimes misaligned scores, occasionally assigning high values to incorrect responses and low values to correct ones. Rule-based rewards (b) enforce a strict binary (0–1) boundary: they rarely give false positives, but due to their stringent criteria, many predictions that are actually correct receive a reward of 0 simply because they fail to pass the rule. HERO (c) uses the rule as a gate, which significantly reduces false positives. At the same time, by integrating the reward model signal, HERO assigns higher reward scores to those cases that would have been false negatives under (b), resulting in more accurate and informative supervision.

## 1 INTRODUCTION

Reasoning lies at the heart of human intelligence, and increasingly, at the frontier of large language model (LLM) capabilities (Jaech et al., 2024; Guo et al., 2025; Zhang et al., 2025b). In tasks such as mathematical problems or generating proofs, reliable reasoning requires models to generate logically

---

*Work done during an internship at Meta.

consistent multi-step solutions that culminate in a verifiably correct outcome. Verifiable rewards implement this idea by running a deterministic checker—such as an exact numeric or string match, a unit test, or a symbolic equivalence check—on a candidate solution $y$ for input $x$. The checker either accepts or rejects the output, producing a sparse but unambiguous signal $r(x, y) \in \{0, 1\}$, which reinforcement learning can propagate through the entire trajectory. Building on this principle, reinforcement learning from verifiable rewards (RLVR) (Chen et al., 2025b) uses these binary signals to train policies toward solutions that pass the checker. Recent systems—including DeepSeek-R1—have advanced this paradigm at scale, leveraging verifier-grounded feedback to improve reasoning (Guo et al., 2025; Zeng et al., 2025; Luo et al., 2025; Yang et al., 2024a).

However, strict 0–1 verification is coarse and brittle: many reasoning tasks allow for partially correct solutions, equivalent answers in alternative formats, or open-ended outputs that resist exact matching. In such cases, symbolic verifiers may under-credit valid solutions (false negatives) or fail to provide any useful signal (Ma et al., 2025; Huang et al., 2025a). Even when applicable, binary rewards induce sparsity: if all rollouts for a prompt receive the same label (all 0s or 1s), group-relative methods such as GRPO (Shao et al., 2024) yield zero relative advantage and thus no useful policy gradient, stalling policy improvement. This brittleness not only reduces sample efficiency but also skews optimization toward easier, strictly verifiable cases–leaving the hardest and most informative prompts underutilized. Our motivating analysis in Section 3.1 further highlights this limitation: on samples where answers are hard to verify, rule-based verifiers frequently fail with correctness. Reward models, in contrast, offer dense supervision by scoring responses on a continuum (Yang et al., 2024b; Liu et al., 2024; Zhang et al., 2025c; Lyu et al., 2025; Liu et al., 2025). Rather than collapsing all incorrect answers into the same category, they capture nuanced quality differences such as partial correctness, clarity of reasoning steps, or proximity to the ground truth. This graded feedback enriches training signals, helping policies learn from partially correct reasoning paths and better allocate credit across diverse rollouts. However, naively combining these dense reward CT model signals with a binary verifier output often destabilizes training. Specifically, when the reward model's continuous signals are naively blended with binary correctness checks, the resulting reward can become noisy or misaligned with the expected semantics of correctness. Figure 1 illustrates this tradeoff: while reward models offer smooth but misaligned signals, rule-based verifiers enforce correctness but lack nuance. Thus, it remains an open question *how to design an effective hybrid framework that preserves the reliability of verifiers while harnessing the richness of reward models.*

To address this challenge, we propose HERO (**H**ybrid **E**nsemble **R**eward **O**ptimization), a reinforcement learning framework that integrates verifier-anchored and dense reward-model signals to provide reliable yet informative supervision. HERO tackles the instability of naive blending through two key innovations. First, it introduces a stratified normalization scheme that bounds reward-model scores within verifier-defined correctness groups. This ensures that dense feedback refines learning only within the set of responses deemed correct by the verifier, preserving correctness guarantees while exploiting nuanced distinctions. Second, HERO employs a variance-aware weighting mechanism that adaptively adjusts the contribution of different prompts during training. Easy prompts, where most responses are uniformly correct or incorrect, contribute little additional learning signal and are down-weighted. In contrast, harder prompts—where candidate responses vary widely and reward-model scores provide valuable discrimination—are emphasized. These components allow HERO to overcome the brittleness of purely binary rewards and the noisiness of dense signals.

We evaluate HERO on diverse math reasoning benchmarks that training with three regimes: training with easy-to-verify samples where exact final-answer checking is possible, hard-to-verify samples with partially correct or format-sensitive solutions, and mixed settings combining both. Across different LLM backbones, HERO consistently outperforms both RM-only and verifier-only baselines, in all three regimes. Notably, on hard-to-verify tasks evaluation based on Qwen-4B-Base, HERO achieves 66.3, which surpasses RM-only training (54.6) by +11.7 points and verifier-only training (57.1) by +9.2 points. Ablations further confirm that anchoring dense signals to verifiable correctness and adaptively reweighting difficult prompts are both critical for stability and efficiency.

## 2 PRELIMINARIES

**Dense reward via reward modeling.** Reward modeling learns a scalar function $r(x, y)$ that evaluates the quality of a response $y$ given a prompt $x$. Based on the Bradley–Terry model (Bradley &

Terry, 1952), the reward function is typically trained on pairwise preference data by minimizing

$$\mathcal{L}_R = -\mathbb{E}_{(x,y_c,y_r)\in\mathcal{D}}[\log \sigma(r(x, y_c) - r(x, y_r))], \tag{1}$$

where $\sigma$ denotes the sigmoid function, $y_c$ is the response that is considered preferred in a comparison, and $y_r$ is the response considered less preferred. Once learned, $r$ can guide reinforcement learning to align the model with human preferences.

**Sparse reward via verifier.** Reinforcement learning with verifiable rewards (RLVR) leverages a deterministic function $r(x, y)$ to assess correctness, assigning a sparse reward (e.g., 1 for correct, 0 for incorrect). All tokens in a response share the same reward, providing unambiguous supervision for tasks with objective ground truth. In mathematical problem solving, the reward function is based on a verifier that checks whether the model's solution matches the ground-truth reference under equivalence transformations. Specifically, a math verifier typically parses the predicted solution into a structured form (e.g., a symbolic expression, final numeric answer, or proof step), simplifies it, and compares it against the reference solution using symbolic algebra tools or logical equivalence checks. The reward function is based on the verifier:

$$\psi(x, y_i, y_{\text{ref}}) = \begin{cases} 1, & \text{if } y_i \text{ is equivalent to } y_{\text{ref}} \text{ given } x, \\ 0, & \text{otherwise.} \end{cases} \tag{2}$$

**Group Relative Policy Optimization.** GRPO (Shao et al., 2024) extends RLVR by optimizing over multiple responses per prompt rather than treating them independently. Instead of relying on a single trajectory, GRPO compares groups of candidate solutions and assigns relative advantages, which stabilizes learning and improves exploration. It also incorporates clipping (as in PPO) to prevent unstable updates and adds a KL penalty to keep the policy close to a reference model. This group-based formulation helps alleviate the gradient sparsity problem of pure verifier rewards and makes optimization more sample-efficient than standard PPO (Yu et al., 2025).

## 3 METHODOLOGY

### 3.1 MOTIVATION: DELVING INTO RULE-BASED VS. RM-BASED VERIFIERS

Building on the preliminaries, we now examine how the two supervision paradigms – rule-based verifiers that provide sparse but precise correctness signals, and reward models that offer dense but potentially noisy preferences – behave on tasks where correctness is difficult to verify. Since the reliability of training hinges on the quality of supervision, understanding their respective strengths and weaknesses is crucial. To this end, we use the `HardVerify_Math` benchmark (Xu et al., 2025), which focuses on challenging verification scenarios. This benchmark consists of 250 hard-to-verify math problems, including 115 manually selected Olympiad questions (He et al., 2024) and 10 MATH test set questions (Hendrycks et al., 2021) that are prone to false negatives due to complex answer formats, as well as 125 Big-Math questions (Albalak et al., 2025) with a Llama-3.1-8B (Dubey et al., 2024) pass rate below 0.05 and identified as difficult by human experts. To ensure a diverse range of response qualities, for each problem we generate answers using three different models: Llama-3.1-8B, Llama-3.3-70B, and Qwen3-8B. This results in a total of 750 responses, which we use to conduct the verifier analysis presented in Table 1.

**Limitations of rule-based verifiers.** To better understand the trade-offs among different verification approaches, we compare several representative verifiers. For rule-based verifiers, we consider `math_reward.py` from the verl library, the `math_verify` module from verl, and the parse and verify functions from the `Math-Verify` library. In addition, we include a generative model-based verifier (`TIGER-Lab/general-verifier` (Ma et al., 2025)), which is specifically trained for chain-of-thought answer verification. This model has demonstrated strong performance and serves as an effective alternative to traditional rule-based methods.

Results in Table 1 highlight clear precision–recall trade-offs. *Function-based rules offer high precision but low recall.* For example, the `math_reward.py` checker is highly conservative: it almost never produces false positives (FPR=0.3%) but fails to recognize many correct answers, resulting in very low recall (10.1%). A more advanced variant, `math_verify.py (in verl)`, achieves the best

balance—near-zero false positives with substantially higher recall. The `math_verify` library extends coverage with normalization heuristics (e.g., handling formatting differences or units) but remains brittle for mismatched orderings such as lists vs. sets, yielding only 38.6% recall.

Table 1: Performance Comparison of Rule-based Verifier, LLM-as-Verifier, and Reward Models.

| Type | Verifier | Recall ↑ | Precision ↑ | FPR ↓ | Acc. ↑ |
|------|----------|----------|-------------|-------|--------|
| Rule-based | `math_reward (verl)` | 10.1 | 97.5 | 0.3 | 53.6 |
| | `math_verify (verl)` | 68.4 | 100.0 | 0.0 | 83.7 |
| | `math_verify (library)` | 38.6 | 96.1 | 1.6 | 67.6 |
| Generative Model-based | `TIGER-Lab/general-verifier` | 49.5 | 89.3 | 6.3 | 70.9 |
| RM-based | AceMath-7B-RM w threshold 1 | 91.7 | 67.7 | 46.4 | 73.2 |
| | AceMath-7B-RM w threshold 3 | 84.2 | 72.7 | 33.5 | 75.6 |
| | AceMath-7B-RM w threshold 5 | 73.8 | 76.6 | 23.9 | 74.9 |
| | AceMath-7B-RM w threshold 7 | 62.4 | 78.5 | 18.1 | 71.9 |

**Reward modeling can generalize to hard-to-verify samples.** We further examine how reward models behave on hard-to-verify samples. Since correctness is directly checkable for verifiable data, most reward models for mathematical reasoning are trained on these verifiable samples (Yang et al., 2024b; Liu et al., 2024; Zhang et al., 2025c; Lyu et al., 2025; Liu et al., 2025). This raises the question: to what extent can such models generalize to tasks where correctness cannot be directly verified?

Here, we investigate this issue by analyzing the performance of a math-focused reward model (AceMath-7B-RM) on the same hard-to-verify tasks. We assess the model using different score thresholds given the scores generated. As shown in Table 1, setting the threshold to 1 (i.e., considering predictions with RM $\geq 1$ as correct) yields a high recall of 91.7% and broad overall coverage, substantially surpassing rule-based verifiers. However, this comes at the cost of lower precision. Increasing the threshold enhances precision but leads to a decrease in recall.

**The need for hybrid reward design.** Our analysis underscores a key tension: *neither rule-based verification nor reward models alone is sufficient*. Purely binary verifiable rewards can be brittle and overly conservative, especially on hard-to-verify samples. This not only reduces sample efficiency but also skews optimization toward easier, strictly verifiable cases—leaving the hardest and most informative prompts underutilized. Reward models, in contrast, offer dense supervision by scoring responses on a continuum and can capture nuanced quality differences such as partial correctness or proximity to the ground truth. These complementary strengths and weaknesses motivate a hybrid approach: anchoring supervision in symbolic verifiers to preserve correctness, while enriching it with the dense signal of reward models to drive effective policy learning. In the next subsection, we describe our proposed approach in detail.

### 3.2 HERO: HYBRID ENSEMBLE REWARD OPTIMIZATION

Motivated by these findings, our design principle is that rule-based rewards should continue to guide the overall reasoning dynamics, while reward models serve as supplementary signals to enrich training. We therefore propose a *hybrid reward framework* that (i) augments binary correctness with dense reward-model scores and (ii) scales supervision according to prompt difficulty. We describe both components in detail below.

**Dense signals anchored to verifiable correctness.** As argued in the motivation, binary verifiers alone provide stable but overly coarse supervision, while reward models offer nuanced distinctions that are easily corrupted if left unconstrained. However, we find that a naive combination of rule-based verification and reward modeling signals tends to undermine the stability of training and render the hybrid approach ineffective (see Appendix A.3). Specifically, when the reward model's

continuous signals are naively blended with binary correctness checks, the resulting reward can become misaligned with the expected semantics of correctness.

To address this, we propose *stratified normalization*, which rescales the continuous scores of the reward model (RM) to align with the range used by traditional binary rule-based methods. Specifically, let $\{r_{\text{rule}}^{(i)}\}_{i=1}^N \subseteq \{0,1\}$ denote the rule-based verifier outputs and $\{r_{\text{RM}}^{(i)}\}_{i=1}^N \subseteq \mathbb{R}$ the corresponding reward-model scores for a group of $N$ rollouts. We first partition the responses according to $r_{\text{rule}}$, and then apply min–max normalization within each group to $r_{\text{RM}}$ resulting in::

$$\hat{r}(x,y) = \begin{cases} -\alpha + 2\alpha \cdot \dfrac{r_{\text{RM}} - \min r_{\text{RM}}}{\max r_{\text{RM}} - \min r_{\text{RM}} + \epsilon}, & r_{\text{rule}} = 0, \\[2ex] (1-\beta) + 2\beta \cdot \dfrac{r_{\text{RM}} - \min r_{\text{RM}}}{\max r_{\text{RM}} - \min r_{\text{RM}} + \epsilon}, & r_{\text{rule}} = 1. \end{cases} \tag{3}$$

Here $\alpha, \beta \in (0,1]$ control the allowable ranges for incorrect and correct groups, with $\epsilon > 0$ preventing division by zero. In practice, we set $\epsilon$ to relatively small value so that the training dynamics are primarily led by rule-based rewards, with the reward model's contribution serving as a supplementary signal. Figure 1 (c) illustrates the effect of our hybrid reward design compared to $r_{\text{RM}}$ (a) and $r_{\text{rule}}$ (b) alone. This design notably differs from traditional pure verifiable rewards, especially for hard-to-verify samples and for groups where all responses are either positive or negative. In such cases, pure rule-based methods do not distinguish between different rollouts, providing no learning signal. As illustrated in Figure 1(c), HERO introduces reward differences within regions where the rule-based rewards are all 0 or all 1, thereby enabling meaningful gradient flow even when the rule-based verifier assigns the same outcome to all rollouts.

The stratified normalization in our hybrid approach is designed to provide the best of both worlds: verifiers ensure the preservation of correctness semantics by constraining the score ranges, while reward models enhance the supervision by introducing gradations within each group. Incorrect responses are clearly distinguished from correct ones, and correct responses are prioritized based on their relative quality. In this manner, dense signals are anchored to symbolic correctness, mitigating the sparsity observed in pure RLVR.

**Variance-aware advantage reweighting.** In the motivation, we argued that not all prompts are equally informative: trivial ones provide little learning signal, while challenging prompts better reveal differences across candidate solutions. A shortcoming of the original GRPO algorithm is that it treats all prompts uniformly, ignoring this variability. The consequence is inefficient use of training capacity—easy prompts dominate optimization even though they provide little additional guidance, while difficult prompts that expose meaningful distinctions are underutilized. To realign training effort with informativeness, we introduce a *variance-aware weighting* scheme. For each prompt, let $\sigma_u$ denote the standard deviation of reward-model scores across candidate responses, with $\bar{\sigma}$ as a running mean. This variance reflects uncertainty: higher values suggest greater disagreement and thus a richer training signal. We define a bounded monotone weighting function:

$$w_{\text{difficulty}}(\sigma_u) = w_{\min} + (w_{\max} - w_{\min}) \cdot \frac{1}{1 + \exp\left(-k(\sigma_u - \bar{\sigma})\right)}, \tag{4}$$

where $w_{\min}$ and $w_{\max}$ set the minimum and maximum weights, and $k$ controls the slope of the transition. In practice, we treat these as tunable hyperparameters; unless otherwise stated, we use $w_{\min} = 0.5$, $w_{\max} = 2.0$, and $k = 5$, ensuring that difficult prompts are up-weighted by at most $2\times$, while trivial prompts retain at least half weight. The final shaped reward is

$$r_{\text{final}}(x,y) = w_{\text{difficulty}}(\sigma_u) \cdot \hat{r}(x,y). \tag{5}$$

This design operationalizes our intuition: ambiguous, high-variance prompts are emphasized because they reveal more about model weaknesses and reward-model sensitivity, while trivial, low-variance prompts are downweighted to avoid wasting capacity. In doing so, the training process not only remains anchored to verifiable correctness through $\hat{r}$, but also allocates learning effort to the most challenging and informative parts of the data.

## 4 EXPERIMENTS

### 4.1 EXPERIMENTAL SETUP

**Training datasets.** A central question is whether reasoning skills acquired through RLVR on verifiable data can generalize to tasks whose correctness cannot be mechanically checked. To empirically examine this, we construct three distinct training datasets based on subsets of the OPENMATH-REASONING (Moshkov et al., 2025) benchmark: (1). easy-to-verify examples, (2). hard-to-verify examples, and (3). a mixture of the two. For easy-to-verify training data, we sample 2,000 problems whose final answers can be deterministically validated using a rule-based *math_verifier (verl)*. For the hard-to-verify-only regime, we likewise sample 2,000 problems from OPENMATHREASONING, where the correct answers have more flexible formats that make rule-based verification challenging (see Appendix A.2.2 for how we filter as well as some qualitative examples). For the mixed set, we combine 1,000 easy-to-verify and 1,000 hard-to-verify problems, allowing the policy to benefit from both robust exact-check supervision and nuanced feedback from unverifiable cases.

**Models.** To evaluate the generalizability of our method across different backbone models, we conduct experiments using the following models: we use Qwen3-4B-Base (Yang et al., 2025) and OctoThinker-8B-Hybrid-Base base (Wang et al., 2025). To stabilize RL training dynamics, we first perform supervised fine-tuning (SFT) on each base model as a cold start (see Appendix A.2.1 for details). All RL experiments are initialized from the same SFT checkpoint.

**Baselines.** As preliminary points of reference, we also report the performance of the base model and a cold-start SFT model. The main baselines are: (1) Reward model, which uses the AceMath-RM-7B reward model (Liu et al., 2024); (2) Rule-based verifier, which relies on binary, rule-based rewards, marking a sample as correct only if the normalized final answer matches the ground truth via `math_verify (library)` in the VERL repo (Sheng et al., 2025). Our method, HERO, is a hybrid approach that combines (1) and (2) into a single reward, making them the most appropriate baselines for comparison. We also compare HERO with a generative model-based verifier—`TIGER-Lab/general-verifier`—and with a large language model (Qwen2.5-7B-Instruct) directly prompted to act as a verifier, as detailed in the Appendix (see Table 7).

**Evaluation.** Since our training data contains both easy-to-verify and hard-to-verify samples, we aim to evaluate whether the model can acquire generalizable reasoning abilities. To this end, we select six test sets: four in which all answers are easy to verify, and two in which the answers are hard to verify.(1). Easy-to-verify testsets includes 4 benchmarks: MATH500 (Hendrycks et al., 2021), AMC (Li et al., 2024), Minerva (Lewkowycz et al., 2022), and Olympiad (He et al., 2024). We report pass@1 averaged over 8 seeds in Table 2. Following (Yang et al., 2024b), we use temperature 0.6 and top-$p$ 0.95, generate $N = 8$ candidates per problem, and evaluate the first decoded output (pass@1). Reported numbers are means over seeds.Correctness is decided by *math_verifier* (normalized numeric/string match with task-specific post-processing).(2). Hard-to-verify testsets: We use temperature 0.6 and top-$p$ 0.95, generate $N = 1$ candidate per problem. Since symbolic checkers cannot reliably provide binary labels for open-ended solutions, we adopt an *LLM-as-a-judge* protocol. Specifically, we use GPT-4o to compare model outputs against ground-truth answers. We evaluate using the HardVerify-Math benchmark (Xu et al., 2025), which consists of 250 samples. Based on the results in Section 3.1, we find that HardVerify-Math is not a particularly challenging filter, as using math_verify yields relatively good results. Therefore, to further evaluate performance on hard-to-verify reasoning tasks, we additionally collect the TextBookReasoning dataset (Fan et al., 2025) (see Appendix A.2.3 for more details).

### 4.2 MAIN RESULTS

**Performance of HERO on the Qwen-based Model** Table 2 shows that HERO consistently outperforms both the reward-model-only and rule-based verifier baselines across all three training data settings: (1) easy-to-verify data, (2) hard-to-verify data, and (3) a mixture of the two. For each training setting, we evaluate on four datasets where the targets are easy-to-verify tasks, as well as two datasets where the targets are hard-to-verify tasks. When trained on easy-to-verify data and evaluated on four easy-to-verify test sets, HERO achieves an average score of 62.0, outperforming

Table 2: **Performance of HERO trained with Qwen3-4B-Base on both easy-to-verify and hard-to-verify reasoning tasks.** The first block reports results on verifiable tasks (MATH500, AMC, Minerva, Olympiad; with average), while the second block presents results on hard-to-verify tasks (HVM, TBR). We compare our approach HERO —which combines two reward signals—with two baselines corresponding to the two signals: AceMath-7B-RM (a reward model) and math_verify (verl), which uses a 0/1 rule as the reward.

| | Easy-to-verify tasks | | | | | Hard-to-verify tasks | | |
|---|---|---|---|---|---|---|---|---|
| | **MATH500** | **AMC** | **Minerva** | **Olympiad** | **Avg. ↑** | **HVM** | **TBR** | **Avg. ↑** |
| Qwen3-4B-Base | 67.5 | 44.1 | 29.4 | 32.1 | 43.3 | 45.2 | 40.2 | 42.7 |
| SFT Cold Start Model | 69.1 | 50.3 | 39.1 | 34.3 | 48.2 | 50.8 | 43.3 | 47.1 |
| **Training with easy-to-verify samples** | | | | | | | | |
| AceMath-7B-RM | 80.2 | 61.6 | 40.6 | 43.3 | 56.4 | 57.2 | 52.0 | 54.6 |
| math_verify (verl) | 82.3 | 61.3 | 44.0 | 45.5 | 58.3 | 61.0 | 53.1 | 57.1 |
| HERO (Ours) | **85.4** | **69.4** | **44.5** | **48.9** | **62.0** | **73.2** | **59.3** | **66.3** |
| **Training with hard-to-verify samples** | | | | | | | | |
| AceMath-7B-RM | 79.6 | 58.8 | 39.9 | 42.1 | 55.1 | 59.2 | 48.2 | 53.7 |
| math_verify (verl) | 76.2 | 46.6 | 28.7 | 38.2 | 47.4 | 58.4 | 50.0 | 54.2 |
| HERO (Ours) | **80.0** | **63.4** | **40.7** | **43.1** | **56.8** | **59.0** | **54.0** | **56.5** |
| **Training with mixed samples** | | | | | | | | |
| AceMath-7B-RM | 79.6 | 58.8 | 39.9 | 42.1 | 55.1 | 58.4 | 49.6 | 54.0 |
| math_verify (verl) | 81.3 | 61.3 | 38.0 | 43.9 | 56.1 | 62.4 | 55.3 | 58.9 |
| HERO (Ours) | **81.6** | **64.4** | **42.1** | **47.0** | **58.8** | **71.4** | **56.7** | **64.1** |

both RM-only (56.4) and rule-based training (58.3). We attribute this improvement to our stratified normalization, which allows hybrid training to exploit both positive and negative correctness groups: while verifier-only training collapses all-correct or all-incorrect batches (yielding zero relative advantage), HERO preserves meaningful gradients within each group through dense intra-group rewards. The advantage of our approach becomes even more pronounced when training on hard-to-verify samples, where rule-based verifiers are brittle and reward models tend to be noisy. Here, HERO attains 56.8, surpassing RM-only (55.1) by +1.7 points and rule-based verifiers (47.4) by a substantial +9.4 points. This improvement is due to anchoring continuous reward-model scores within verifier correctness groups, which prevents reward drift and ensures stable learning even when binary labels saturate. By combining the precision of rule-based verifiers with the smooth discrimination of reward models, HERO is able to leverage partially correct reasoning paths that would otherwise be discarded by rule-based systems, thereby improving both stability and coverage. When trained on mixed data, which combines easy-to-verify and hard-to-verify samples, HERO again achieves the average (58.8), outperforming RM-only (55.1) and rule-based verifier (56.1) on verifiable tasks.

The advantage becomes even clearer on hard-to-verify evaluations (HVM, TBR), where rule-based verifiers fail to capture partial correctness and reward models are prone to drift. Here, HERO attains 66.3 when trained on easy-to-verify data, outperforming RM-only (54.6) by +11.7 points and rule-based training (57.1) by +9.2 points. When trained on hard-to-verify samples, it still leads with 56.5 compared to 53.7 (RM-only) and 54.2 (rule-based). Under mixed training, HERO reaches 64.1, surpassing both baselines by large margins on hard-to-verify tasks. These results highlight that hybrid reward design generalizes robustly across both verifiable and hard-to-verify tasks, yielding stable improvements regardless of whether evaluation relies on symbolic checking or model judgment. Overall, hybrid reward learning delivers consistent improvements across all settings, demonstrating that structured reward integration is critical for reasoning tasks that go beyond strict verifiability. We note that the magnitude of gains in Table 2 naturally varies across training regimes due to differences in reward quality, rather than instability of HERO. With easy-to-verify training data, the verifier is accurate and often positive, so HERO can fully exploit its hybrid design and achieve large improvements; with hard-to-verify data, the verifier rarely fires and many prompts receive all-0 labels, so the learning signal is weaker and gains are necessarily smaller. In the mixed regime, easy-to-verify samples provide a strong verifier anchor while hard-to-verify samples reduce the domain gap to difficult test sets, which explains why improvements are modest on verifiable tasks but large again on hard-to-verify evaluations.

Table 3: **Performance of HERO trained with OctoThinker-8B-Hybrid-Base on both easy-to-verify and hard-to-verify reasoning tasks.** The first block shows results on four easy-to-verify tasks, which reported pass@1 averaged over 8 seeds. The second block show results on two hard-to-verify testsets, which reported GPT4.1 judges results.

| | Easy-to-verify tasks | | | | | Hard-to-verify tasks | | |
| --- | --- | --- | --- | --- | --- | --- | --- | --- |
| | MATH500 | AMC | Minerva | Olympiad | Avg. ↑ | HVM | TBR | Avg. ↑ |
| OctoThinker-8B-Hybrid-Base | 32.0 | 15.3 | 9.1 | 11.0 | 16.9 | 26.0 | 21.1 | 23.6 |
| SFT Cold Start Model | 56.0 | 35.9 | 19.7 | 21.6 | 33.3 | 27.6 | 26.4 | 27.0 |
| **Training with easy-to-verify samples** | | | | | | | | |
| AceMath-7B-RM | 62.3 | 38.4 | 26.2 | 25.5 | 38.1 | 29.6 | 27.8 | 28.7 |
| math_verify (verl) | 60.1 | 39.4 | 26.7 | 24.1 | 37.6 | **31.6** | 28.9 | 30.3 |
| HERO (Ours) | **63.0** | **40.6** | **30.1** | **26.7** | **40.1** | 28.4 | **36.7** | **32.6** |
| **Training with hard-to-verify samples** | | | | | | | | |
| AceMath-7B-RM | 60.7 | 33.8 | 22.4 | 24.9 | 35.4 | 32.0 | 29.8 | 30.9 |
| math_verify (verl) | 60.0 | 29.7 | 23.9 | 24.8 | 34.6 | 28.8 | 26.7 | 27.8 |
| HERO (Ours) | **64.9** | **41.6** | **27.9** | **29.6** | **41.0** | **32.4** | **36.7** | **34.6** |
| **Training with mixed samples** | | | | | | | | |
| AceMath-7B-RM | 60.2 | 34.4 | 24.0 | 23.8 | 35.6 | 30.8 | 29.3 | 30.1 |
| math_verify (verl) | 59.3 | 33.7 | 24.7 | 24.0 | 35.4 | 27.6 | 28.7 | 28.2 |
| HERO (Ours) | **65.2** | **38.1** | **28.1** | **29.3** | **40.2** | **34.8** | **31.6** | **33.2** |

**Performance of HERO on the OctoThinker-based Model**    On Qwen3-4B-Base (Table 2), which already shows strong performance, HERO consistently delivers clear improvements across all evaluation settings. On OctoThinker-8B-Hybrid-Base (Table 3), which starts from a much weaker baseline of 16.9 on verifiable and 23.6 on hard-to-verify tasks, HERO achieves substantial absolute and relative gains. When trained on easy-to-verify samples, it reaches 40.1 on verifiable and 32.6 on hard-to-verify evaluations, surpassing both the reward-model baseline (38.1/28.7) and the rule-based verifier (37.6/30.3). Training on hard-to-verify samples yields similar improvements, achieving 41.0/34.6 compared to 35.4/30.9 (RM-only) and 34.6/27.8 (verifier-only). Training on mixed training samples, it maintains the highest averages of 40.2 and 33.2, outperforming all baselines by 4–6 points. These results show that hybrid reward design generalizes robustly across model scales—preserving the verifier's stability for stronger models like Qwen3-4B-Base while bringing large relative gains to weaker ones such as OctoThinker-8B-Hybrid-Base.

**Verifier-only training struggles on hard-to-verify tasks.**    Symbolic verifiers, while precise, perform poorly on open-ended or format-sensitive reasoning. On Qwen3-4B-Base (Table 2), verifier-only training reaches only 47.4 on hard-to-verify tasks—worse than HERO (56.5), RM-only (53.7), and even slightly below the SFT baseline (47.1). Similar degradation appears on OctoThinker-8B-Hybrid-Base (Table 3), where verifier-only supervision lags far behind HERO. The core issue is that weaker models often produce rollouts that are uniformly labeled 0 or 1, causing group-relative objectives such as GRPO to yield zero gradients. In contrast, HERO adds dense intra-group variation via reward-model refinement, preserving gradient flow even when binary labels saturate and allowing the policy to separate partially correct from entirely incorrect solutions.

## 4.3    ADDITIONAL ABLATIONS

**Dense negative ranges are more important than positive samples.**    We evaluate the role of dense negative and positive reward on the setting of training with easy-to-verify samples based on Qwen-4B-Base. We found dense reward in the negative group play a more criticak role in stabilizing training and improving learning efficiency than dense reward in the positive group devided by HERO.. While positives signal correctness, negatives provide richer supervision by penalizing diverse reasoning errors. Notably, dense negative rewards but maintaining sparse verifier positive rewards boosts performance on verifiable tasks from 59.4 to 61.4, and even more on hard-to-verify tasks from 62.2 to 68.4 (see Figure 2). This demonstrates that well-calibrated negative ranges are essential: they provide broader feedback, enabling the model to detect subtle errors and generalize better on complex cases.

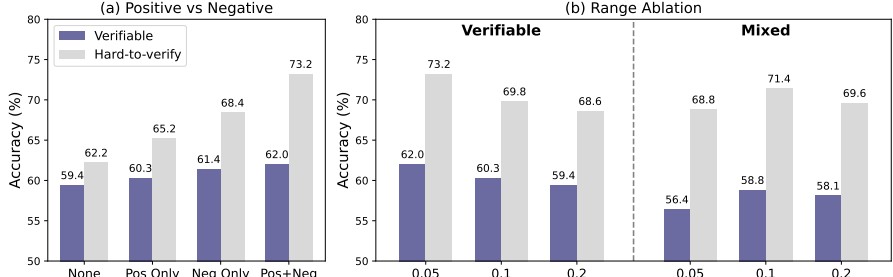

Figure 2: **(a) Impact of using positive and negative dense ranges.** Dense negative rewards contribute more to stable learning than positive samples. **(b) Effect of varying reward ranges under different training regimes.** Smaller ranges perform best on verifiable tasks, while larger ranges benefit mixed settings by providing denser feedback.

**Reward range selection is crucial for balancing stability and performance.** We conducted ablation studies to investigate the impact of varying reward ranges on model performance by training with easy-to-verify samples based on Qwen-4B-Base, as shown in Figure 2(b). For verifiable tasks, smaller reward ranges (e.g., $\alpha = 0.05$) yielded the best results, as the rule-based verifier's precision benefits from a tighter range that minimizes noise and maintains stability. Expanding the range beyond this threshold led to diminishing returns and increased noise. In contrast, for mixed tasks, where many samples fail the rule-based verifier, the learned reward model plays a larger role. Here, larger reward ranges (e.g., $\alpha = 0.1$ or $\alpha = 0.2$) provided richer signals, allowing the model to learn more effectively from harder tasks. However, expanding the range beyond a certain point caused a slight performance drop due to overfitting or excessive noise. Overall, careful tuning of the reward range, particularly for the negative rewards, is crucial to balancing stability and performance: *datasets with relatively dense informative rewards and few all-positive/all-negative groups tend to benefit from smaller ranges, whereas datasets with many all-positive/all-negative groups are better served by slightly larger ranges that inject more intra-group variation.*

**Variance-aware reweighting in HERO improves the model's reasoning ability.** We evaluated variance-aware reweighting based on reward-model score variance, which emphasizes ambiguous, high-variance prompts while down-weighting trivial ones to reduce overfitting with the setting of training with easy-to-verify sam-

Table 4: Variance-aware reweighting improves performance on both verifiable and hard-to-verify samples.

| Methods | Easy-to-verify | Hard-to-verify |
|---|---|---|
| w/o reweighting | 60.8 | 69.4 |
| w reweighting | **62.0** | **73.2** |

ples based on Qwen-4B-Base. This dynamic adjustment yields consistent gains, particularly on hard-to-verify tasks where dense signals are most informative. As shown in Table 4, reweighting improves accuracy on both verifiable and hard-to-verify benchmarks, with larger gains in the latter (+3.8), confirming that focusing capacity on uncertain samples leads to more robust and generalizable improvements.

## 5 RELATED WORK

**Reinforcement learning from verifiable rewards.** Reinforcement learning from verifiable rewards (RLVR) leverages deterministic correctness checks—such as passing unit tests or matching reference answers—to enhance learning (Shao et al., 2024). Early program synthesis work demonstrated that agent-generated trajectories validated against ground truth outperform supervised approaches (Bunel et al., 2018; Chen et al., 2021). For LLMs, rule-based verification plays a crucial role in filtering, providing training signals, and supporting benchmark evaluations (Xiong et al., 2025; Yu et al., 2025; Shao et al., 2024). Recent extensions include: outcome-driven RL (GRPO) for grounding and rubric-anchored RL which introduces structured rubrics for open-ended response evaluation (Huang et al., 2025b); verifier-free RL strategies like VeriFree, which bypass explicit checking by directly maximizing the probability of generating the reference answer while achieving performance on par with verifier-based methods (Zhou et al., 2025); and cross-domain RLVR,

which employs LLM-derived scoring for domains lacking reference answers (Su et al., 2025). Despite these advances, function-based rule verifiers remain high-precision but low-recall: they often assign zero reward to semantically correct yet textually divergent outputs (Huang et al., 2025a), which has motivated the use of learned, model-based verifiers (Huang et al., 2025a; Chen et al., 2025a; Ma et al., 2025; Xu et al., 2025). However, the coverage and generalization of LLM-based verifiers are still limited (Li et al., 2025), and in many existing "hybrid" schemes they are invoked only to relabel a subset of failures, ultimately producing binary 0/1 signals. As a result, these approaches continue to suffer from sparse outcome-level rewards and the classical RLVR issue that all-positive or all-negative rollouts yield vanishing advantages, limiting data efficiency. In contrast to previous work, we propose a hybrid approach that combines rule-based verification with continuous, dense reward signals from learned models, allowing us to maintain the stability of verifiers while addressing their sparsity. By anchoring dense signals to symbolic correctness and introducing a variance-aware weighting mechanism, our method enables more informative, stable, and sample-efficient learning on both verifiable and hard-to-verify tasks.

**Reasoning on hard-to-verify tasks.** As the reasoning capabilities of LLMs have reached new heights, increasingly challenging reasoning benchmarks have been proposed (Phan et al., 2025; Zhang et al., 2025a). These problems often involve complex outputs, such as natural language representations and intricate mathematical or physical formulas. In such cases, rule-based verification methods, while effective for well-defined problems, struggle to capture the nuances of these tasks. Recent work focused on the LLMs as judges, where LLMs assess the quality of generated responses (Chen et al., 2025a; Ma et al., 2025; Huang et al., 2025a; Xu et al., 2025; Li et al., 2025), enabling more nuanced evaluations. However, despite its conceptual simplicity, LLM-as-judge may not always produce reliable assessments for domain-specific or long-form data. Some recent work proposes going beyond binary labels from verifiers for hard-to-verify tasks. For example, Gurung & Lapata (2025) applies reasoning traces in Next-Chapter Prediction (NCP) for long-form story generation via likelihood estimation, while Tang et al. (2025) uses Jensen's evidence lower bound to treat chain-of-thought reasoning steps as latent variables in the generative process. They directly discard the verifier component. In contrast, our work retains the use of verifiable rewards but enhances supervision through the introduction of a reward model. Peng et al. (2025) directly add the verifiable correctness signals and the human preferences for agentic tasks.

## 6 CONCLUSION

We introduced HERO (**H**ybrid **E**nsemble **R**eward **O**ptimization), which combines a rule-based verifier $r_{\text{rule}} \in \{0, 1\}$ with a dense reward model via stratified normalization and variance-aware weighting. By anchoring reward-model scores to verifier-defined correctness groups and emphasizing informative prompts, HERO preserves the precision and stability of verifiers while supplying dense, trajectory-sensitive feedback, mitigating gradient sparsity and RM-only drift. Empirically, HERO consistently outperforms RM-only and verifier-only baselines across verifiable, hard-to-verify, and mixed regimes and across two backbones, showing that structured hybrid reward design is effective for math reasoning. HERO is a first step toward more general hybrid reward frameworks: it currently relies on reasonably accurate rule-based signals in math domains, and extends naturally to richer verifiers, process-level supervision, and adaptive weighting schemes. We hope these results and analyses provide a useful foundation for future work on combining symbolic and learned feedback for reasoning beyond strictly verifiable settings.

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

# Appendix

## A EXPERIMENTS

### A.1 EXPERIMENTAL SETUP

| Category | Hyperparameter | Value |
|---|---|---|
| Data | Train file | OPENMATHREASONING |
| | Max prompt length | 1024 |
| | Max response length | 8192 |
| | Filter overlong prompts | True |
| Actor Model | Base model 1 | Qwen3-4B-Base |
| | LR | $1 \times 10^{-6}$ |
| | KL loss coefficient $\beta$ | 0 |
| | Entropy loss | 0 |
| | Use dynamic batch size | True |
| Rollout | Rollout engine | vllm |
| | GPU mem utilization | 0.6 |
| | Train rollout n | 8 |
| | Temperature | 1.0 |
| Reward | Rule Based | Math_Verify |
| | Reward Model Based | AceMath-RM-7B |
| Trainer | Mini Batch size | 128 |
| | Full Batch size | 512 (4 step off-policy) |
| | Critic Warmup | 0 |
| | GPUs/node | 4 |
| | Nodes | 8 |
| | Total epochs | 20 |
| | Clip Ratio | (0.2, 0.28) |

Table 5: Key hyperparameters used for GRPO training on OPENMATHREASONING (Moshkov et al., 2025) in the verl (Sheng et al., 2025) framework for the Qwen-4B-Base.

| Category | Hyperparameter | Value |
|---|---|---|
| Data | Train file | OPENMATHREASONING |
| | Max prompt length | 1024 |
| | Max response length | 4096 |
| | Filter overlong prompts | True |
| Actor Model | Base model 1 | `OctoThinker-8B-Hybrid-Base` |
| | LR | $1 \times 10^{-6}$ |
| | KL loss coefficient $\beta$ | 0.001 |
| | Entropy loss | 0 |
| | Use dynamic batch size | True |
| Rollout | Rollout engine | vllm |
| | GPU mem utilization | 0.6 |
| | Train rollout n | 16 |
| | Temperature | 1.0 |
| Reward | Rule Based | `Math_Verify` |
| | Reward Model Based | `AceMath-RM-7B` |
| Trainer | Mini Batch size | 128 |
| | Full Batch size | 512 (4 step off-policy) |
| | Critic Warmup | 0 |
| | GPUs/node | 4 |
| | Nodes | 8 |
| | Total epochs | 20 |
| | Clip Ratio | (0.2, 0.28) |

Table 6: Key hyperparameters used for GRPO training on OPENMATHREASONING (Moshkov et al., 2025) in the verl (Sheng et al., 2025) framework for the OctoThinker-8B-Hybrid-Base.

**HERO hyper-parameters.** For hybrid reward training for both Qwen-4B-Base and OctoThinker-8B-Hybrid-Base, we set the range parameters $\alpha$ and $\beta$ depending on the task type. For easy-to-verify tasks, we adopt a tighter setting $\alpha = \beta = 0.05$ to exploit the high precision of rule-based verifiers while minimizing noise. For mixed and hard-to-verify tasks, where the reward model contributes more substantially to supervision, we relax the range to $\alpha = \beta = 0.1$ to provide richer feedback. For variance-aware reweighting, we fix the weighting bounds as $w_{\min} = 0.4$ and $w_{\max} = 3.0$, with a steepness parameter $k = 6$ in the logistic weighting function. These values ensure that trivial prompts are down-weighted, while highly uncertain prompts—where reward-model scores vary widely—receive stronger emphasis without destabilizing training.

**Training hyper-parameters.** Tables 5 and 6 provide an overview of the hyperparameter configurations used in our GRPO training runs with Qwen3-4B-Base and OctoThinker-8B-Base. The tables cover settings across data preparation, actor model optimization, rollout generation, reward specification, and trainer configuration. They highlight the consistent use of OPENMATHREASONING as the training corpus, the integration of both rule-based and reward-model signals, and the adoption of scalable rollout and training strategies within the verl framework. Together, these summaries document the experimental setup and ensure reproducibility across different backbone models. In addition, we employ the HuggingFace `math_verify` library to provide standardized rule-based verification of responses against ground-truth answers, which guarantees consistency in supervision across all experiments.

**Evaluation details.** For easy-to-verify test sets, we follow Yang et al. (2024b): we use temperature 0.6 and top-$p = 0.95$, generate $N = 8$ candidates per problem, and report pass@1 (first decoded output) averaged over 8 seeds (Table 2). Correctness is determined by *math_verifier* (normalized numeric/string match with task-specific post-processing). For hard-to-verify test sets, we use the

same temperature and top-$p$ but generate $N = 1$ sample per problem, and rely on GPT-4o as a judge to compare model outputs with ground-truth answers. HardVerify-Math (Xu et al., 2025) contains 250 samples and, as discussed in Section 3.1, is not a particularly strict filter, since *math_verifier* already achieves relatively good performance. To further stress-test hard-to-verify reasoning, we additionally evaluate on the TextBookReasoning dataset (Fan et al., 2025); see Appendix A.2.3 for construction details.

## A.2    DATA PREPARATION

### A.2.1    SUPERVISED FINE-TUNING DATASET PREPARATION

We found that initiating RL training directly from the base model often resulted in instability, particularly in the absence of a cold start. For instance, the Qwen3-4B-Base model frequently produced mixed-language outputs and generated irrelevant content during the early stages of training. Similarly, the octothinker base model demonstrated multi-turn behavior, leading to highly variable response lengths. To mitigate these issues and enhance the stability of RL training, we first conducted two epochs of cold-start supervised fine-tuning (SFT) before beginning RL. To avoid unintentional distillation from more capable models, we used the base model itself to generate responses. These outputs were then filtered, retaining only samples that satisfied the following criteria: the response contained the correct final answer, was entirely in English, and did not exhibit any unstop issues. For cold start training, we ultimately used only 2,000 SFT samples.

### A.2.2    TRAINING DATA FILTER FROM OPENMATHREASONING

In this paper, we focus on reasoning questions that have extractable answers. To this end, we exclusively utilize data from the OpenMathReasoning dataset, selecting only those examples where the problem_type is set to has_answer_extracted. From the CoT split, we extracted 40k examples. For each example, we generated solutions and extracted the predicted answers, which were then verified using math_verifier (verl). We randomly sampled 2k examples that passed the verifier to serve as verifiable training data, and another 2k examples that failed verification as hard-to-verify training samples. These two sets were combined to create a mixed training dataset for reinforcement learning (RL) training. We use math_verifier (verl) to filter all the samples.

### A.2.3    HARD-TO-VERIFY EVALUATION BENCHMARK FROM TEXTBOOKREASONING

```
GPT-4o filter prompt for TextBookReasoning.

"I am looking for math questions that are suitable for evaluating a math model. Please
    help me select questions that meet the following criteria:

1. The question must be clear and unambiguous.

2. The question must have a specific, factual, and answerable solution (not open-ended or
    subjective).

3. The question must NOT require a proof or explanation of reasoning.

4. The question must NOT be a statement; it should be a direct question.

For each question I provide, please respond with:
- \"Conclusion: Suitable\" in the end if the question meets all the criteria above.

- \"Conclusion: Not Suitable\"

If the question does not meet the criteria, briefly explain why."
```

Figure 3: GPT-4o filter prompt for TextBookReasoning.

To construct a more challenging and reliable benchmark for hard-to-verify tasks, we employ the **TextBookReasoning** benchmark. The following criteria were used to filter and refine the dataset for the evaluation:

1. **Pass-through Math Verification Filter**
   The initial step in filtering was to ensure that the answers in the dataset did not pass the `math_verify` check, ensuring that the questions and answers involved a certain level of complexity or ambiguity that would make them challenging for standard verifiers.

2. **Llama 3.3_70B Instruct Model for Natural Reasoning**
   The dataset was further refined by using the `Llama 3.3_70b_instruct` model to answer natural reasoning prompts. Only the prompts for which Llama could not provide an answer were kept for further evaluation. This step ensured that the dataset included questions that required more advanced reasoning abilities, beyond the capabilities of standard models.

3. **GPT-4 as the Final Filter**
   Finally, GPT-4 was used to filter out questions that still met the criteria of being complex and hard-to-verify. GPT-4's ability to handle nuanced reasoning ensured that only the most challenging prompts remained. The prompt is shown as Figure 3

This process ultimately resulted in a refined set of approximately **750** prompts suitable for hard-to-verify task evaluation.

**Prompt Template for Hard-to-Verify Tasks Evaluation**    The evaluation of student answers to these prompts is based on the following template, which uses GPT-4 to compare the student's answer against the ground truth:

**Math Question Selection Criteria**    The following prompt was used to select math questions suitable for evaluating a math model. The criteria for question selection are outlined below:

### A.2.4    HARD-TO-VERIFY PROMPT

We set the hard-to-verify evaluation prompt as shown in Figure 4. This template is designed to assess whether a student's response matches the reference answer without re-solving the question. By explicitly instructing GPT-4o to perform equivalence checking rather than problem solving, the protocol minimizes leakage of additional reasoning and focuses purely on correctness judgment. The structured format, including the question, ground truth, and student answer, ensures consistency across evaluations and reduces prompt sensitivity, making it suitable for benchmarking performance on hard-to-verify tasks.

```
Prompt Template for hard-to-verify tasks evaluation via GPT-4o.

User: ### Question: {question}

### Ground Truth Answer: {ground_truth}

### Student Answer: {student_answer}

For the above question, please verify if the student's answer is equivalent to the ground
    truth answer.
Do not solve the question by yourself; just check if the student's answer is equivalent to
     the ground truth answer.
If the student's answer is correct, output "Final Decision: Yes". If the student's answer
    is incorrect, output "Final Decision: No".

Assistant:
```

Figure 4: Prompt Template for hard-to-verify tasks evaluation via GPT-4o.

Table 7: **Comparison with model-based verifiers on Qwen-3-4B-Base.** Results are pass@1 averaged over 8 seeds across verifiable and hard-to-verify reasoning tasks. HERO consistently outperforms both General Reasoner and Qwen2.5-7B-Instruct under all regimes.

| | Easy-to-verify tasks | | | | | Hard-to-verify tasks | | |
|---|---|---|---|---|---|---|---|---|
| | **MATH500** | **AMC** | **Minerva** | **Olympiad** | **Avg. ↑** | **HVM** | **TBR** | **Avg. ↑** |
| **Training with easy-to-verify samples** | | | | | | | | |
| *General Reasoner* | 82.8 | 62.8 | 43.8 | 45.0 | 58.6 | 62.8 | 54.0 | 58.4 |
| *Qwen2.5-7B-Instruct* | 83.7 | 58.1 | 43.1 | 47.4 | 58.1 | 68.0 | 57.1 | 62.5 |
| HERO *(Ours)* | 85.4 | 69.4 | 44.5 | 48.9 | 62.0 | 73.2 | 59.3 | 66.3 |
| **Training with hard-to-verify samples** | | | | | | | | |
| *General Reasoner* | 78.6 | 56.3 | 38.7 | 41.5 | 53.8 | 59.6 | 48.4 | 54.0 |
| *Qwen2.5-7B-Instruct* | 78.2 | 60.5 | 41.8 | 41.7 | 55.6 | 57.2 | 51.7 | 54.5 |
| HERO *(Ours)* | 80.0 | 63.4 | 40.7 | 43.1 | 56.8 | 59.0 | 54.0 | 56.5 |
| **Training with mixed samples** | | | | | | | | |
| *General Reasoner* | 81.4 | 61.2 | 43.2 | 46.5 | 58.1 | 64.0 | 54.0 | 59.0 |
| *Qwen2.5-7B-Instruct* | 80.4 | 63.1 | 40.5 | 48.0 | 58.0 | 68.8 | 57.7 | 63.3 |
| HERO *(Ours)* | 81.6 | 64.4 | 42.1 | 47.0 | 58.8 | 71.4 | 56.7 | 64.1 |

## A.3 MORE EXPERIMENTS

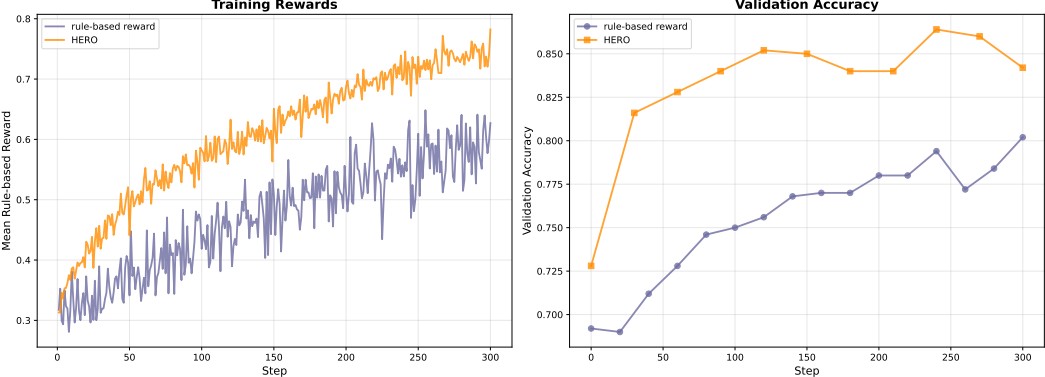

Figure 5: **RL training curves on MATH500 (easy-to-verify training).** Left: mean rule-based reward computed by math_verifier. Right: validation accuracy on MATH500.

**RL training curves comparison between HERO and baseline.** Figure 5 compares the RL dynamics of the rule-based baseline and HERO when both are trained only on easy-to-verify samples and evaluated on MATH500. On the left, we plot the mean rule-based reward (computed by math_verifier) over training steps. The rule-based baseline starts with a relatively low reward and increases slowly, ending around 0.63 after 300 steps. In contrast, HERO climbs much faster and reaches a substantially higher plateau (around 0.75–0.78), consistently staying above the baseline throughout training. On the right, we show the corresponding validation accuracy on MATH500. The rule-based baseline improves from 0.692 at step 0 to 0.802 at step 300, following a gradual upward trend. HERO starts slightly higher at 0.728, quickly jumps above 0.80 by step 30, peaks around 0.852 at step 120, and then fluctuates in the 0.84–0.86 range (e.g., 0.864 at step 240 and 0.842 at step 300). Overall, HERO not only achieves a higher final accuracy, but also maintains a persistent 4–8 point advantage over the rule-based baseline across most of training, indicating that the hybrid reward improves both convergence speed and the entire training trajectory on easy-to-verify data.

**Hybrid reward surpasses model-based verifiers across all the three regimes.** To further assess whether hybrid reward learning can outperform existing model-based verifiers, we compare HERO against two representative systems: General Reasoner, a frozen 1.5B verifier model that provides binary correctness judgments (Ma et al., 2025), and Qwen2.5-7B-Instruct , a large instruction-tuned

verifier (Yang et al., 2024b). As shown in Table 2, HERO consistently achieves higher accuracy than both model-based verifiers under all training regimes. When trained with verifiable samples, HERO attains an average score of 62.0, outperforming General Reasoner (58.4) and Qwen2.5-7B-Instruct (62.5) while maintaining greater stability across datasets such as MATH500 (85.4 versus 82.8 and 83.7) and AMC (69.4 versus 62.8 and 58.1). In the hard-to-verify regime, the advantage becomes more pronounced: HERO reaches 56.5, exceeding General Reasoner (54.0) and Qwen2.5-7B-Instruct (54.5), demonstrating that hybrid reward learning provides more reliable supervision even when symbolic verification is unreliable and model-based signals are uncertain. In the mixed setting, which combines both verifiable and open-ended samples, HERO again leads with 64.1, surpassing General Reasoner (59.0) and Qwen2.5-7B-Instruct (63.3). These results highlight that integrating verifier-anchored and reward-model signals yields not only better accuracy but also more consistent generalization across regimes, outperforming larger model-based verifiers despite using no additional model parameters or external training data. The improvement underscores that structured reward integration, rather than sheer verifier scale, is the key to effective and robust reasoning optimization.

**The proposed method does not rely on large reward models.** A natural question is whether stronger supervision requires scaling up the reward model itself. To isolate this factor, we replace the 7B reward model in HERO with a much larger 72B reward model, keeping the verifier and all training configurations fixed. As shown in Table 8, the larger reward model

Table 8: Impact of reward model size: a larger RM provides in HERO no remarkable gain over the HERO with smaller RM .

| Reward model | Easy-to-verify | Hard-to-verify |
|---|---|---|
| AceMath-RM-7B | 62.0 | 73.2 |
| AceMath-RM-72B | 62.8 | 71.4 |

yields only a marginal improvement on verifiable tasks (62.8 vs. 62.0) and even slightly underperforms on hard-to-verify tasks (71.4 vs. 73.2). This confirms that the gains of HERO primarily come from its hybrid reward formulation—through stratified normalization and variance-aware weighting—rather than from reward model scaling. Practically, this means that HERO can achieve strong results with compact reward models, offering better efficiency and deployability without sacrificing accuracy.

**Naively combining rule-based rewards and reward signals from reward modeling does not perform well.** A direct integration of rule-based verification and reward signals from reward modeling, without proper structural alignment, often disrupts the stability of training. As shown in Table 9, when the weight of the rule-based reward is varied ($\alpha = 0.1$, 0.5, and 0.9), the combined reward performance remains suboptimal, with scores ranging from 55.9 to 58.7 for verifiable tasks and 60.2 to 61.4 for hard-to-verify tasks.

Table 9: $\alpha$ represents the weight of the rule-based reward.

| Methods | Easy-to-verify | Hard-to-verify |
|---|---|---|
| Reward combine ($\alpha$=0.1) | 57.6 | 60.2 |
| Reward combine ($\alpha$=0.5) | 58.7 | 61.4 |
| Reward combine ($\alpha$=0.9) | 55.9 | 60.4 |
| HERO (Ours) | 62.0 | 73.2 |

Specifically, when the continuous signals from the reward model are naively combined with binary correctness checks, the resulting reward can become noisy or misaligned with the intended notion of correctness. Without explicitly constraining the continuous scores within the rigid framework of the verifier's correctness criteria, reward-model outputs can be distorted by imperfections in the model, diminishing both interpretability and precision in the feedback. Moreover, the lack of a safeguard to differentiate true positives from noisy results can lead the model to exploit unintended patterns, which may not align with human expectations. As a result, an unrefined fusion of these two reward signals can dilute the benefits of both approaches, destabilizing the learning process.

**Reward models hack faster on hard-to-verify samples.** Since the reward model (RM) is trained on outcome-based verifiable samples (Liu et al., 2024), it is important to examine its behavior across datasets with varying levels of verifiability. We evaluate four datasets: DAPO (Yu et al., 2025), which is easy to verify; OpenMath Verifiable, which passes the math_verifier; OpenMath Non-

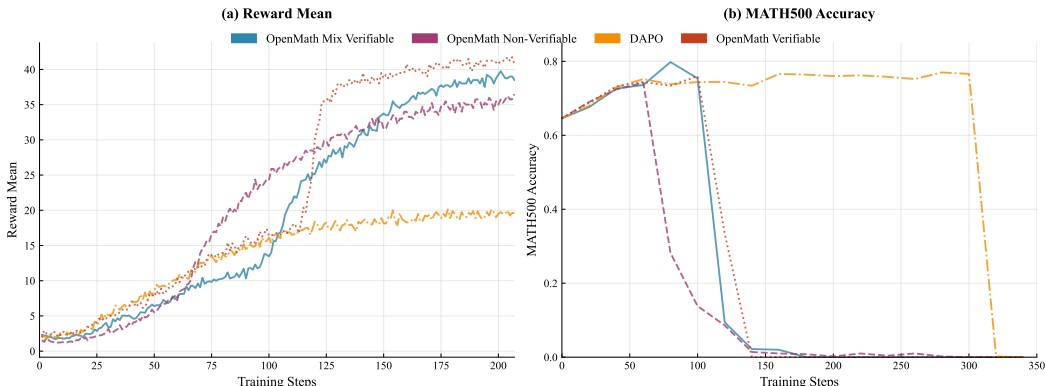

Figure 6: Reward model qualification ability on mixed groups: (a) distribution of AUROC scores, (b) AUROC box plot, (c) cumulative distribution of AUROC, and (d) AUROC performance categories.

Verifiable, which is harder to verify; and OpenMath Mix Verifiable, which combines both. As shown in Figure 6, the RM rapidly increases the reward mean across all datasets, with the sharpest gains on OpenMath Non-Verifiable and OpenMath Mix Verifiable. For example, on Non-Verifiable data, the reward mean climbs steeply from below 5 to over 30 within the first 100 training steps, and peaks above 40 by step 150. However, MATH500 accuracy collapses shortly after, dropping from around 0.75 at step 50 to below 0.2 by step 100, and effectively to zero by step 150. A similar trend appears on Mix Verifiable: accuracy initially rises to about 0.8 at step 100 but then crashes to nearly zero by step 150, despite the reward mean continuing to rise steadily past 35. In contrast, OpenMath Verifiable shows slower but steadier progress: rewards grow more gradually, and accuracy improves to about 0.8 by step 120 before stabilizing without collapse. DAPO also exhibits stable optimization, with accuracy consistently around 0.75–0.78 as rewards increase moderately. These results highlight a clear mismatch: rapid reward gains on hard-to-verify tasks are not evidence of genuine reasoning improvement, but rather reward hacking that leads to catastrophic accuracy collapse. This illustrates the brittleness of relying solely on dense reward models and motivates hybrid reward frameworks that combine verifier-anchored reliability with the nuance of dense signals.

## B QUALITATIVE ANALYSIS

### B.1 REWARD MODEL QUALIFICATION ABILITY

To better understand the reliability of reward-model supervision, we analyze its ability to approximate the verifier signal as a binary classification task. We randomly take all the rollouts from one step (the 250 for the verifiable samples training) during the training. Specifically, we treat the reward model's raw scores as logits and the verifier's outputs as ground-truth binary labels, then compute AUROC statistics to measure discriminative power.

Figure 7 shows four complementary views. The histogram (top-left) reveals a strong skew toward high AUROC values, with a mean of 0.79 and a median of 0.92, indicating that the reward model often ranks correct responses above incorrect ones. The box plot (top-right) highlights robustness but also exposes several low outliers where the model fails to separate classes. The cumulative distribution (bottom-left) confirms that roughly 80% of groups achieve AUROC above 0.7. Finally, the performance categorization (bottom-right) shows that 56.8% of groups reach "excellent" AUROC ($\geq 0.9$), while only 13.7% fall into the "random/poor" range (0.4–0.6).

These results suggest that although the reward model is not perfect, it provides reliable ranking signals in the majority of cases. Importantly, this supports the use of dense reward signals to refine learning within verifier-defined groups: while the verifier anchors correctness, the reward model adds discriminative power that helps differentiate among responses of varying quality. The presence of failure cases further justifies our hybrid framework, which uses stratified normalization to bound reward-model signals within verifier groups, ensuring stability even when AUROC is low.

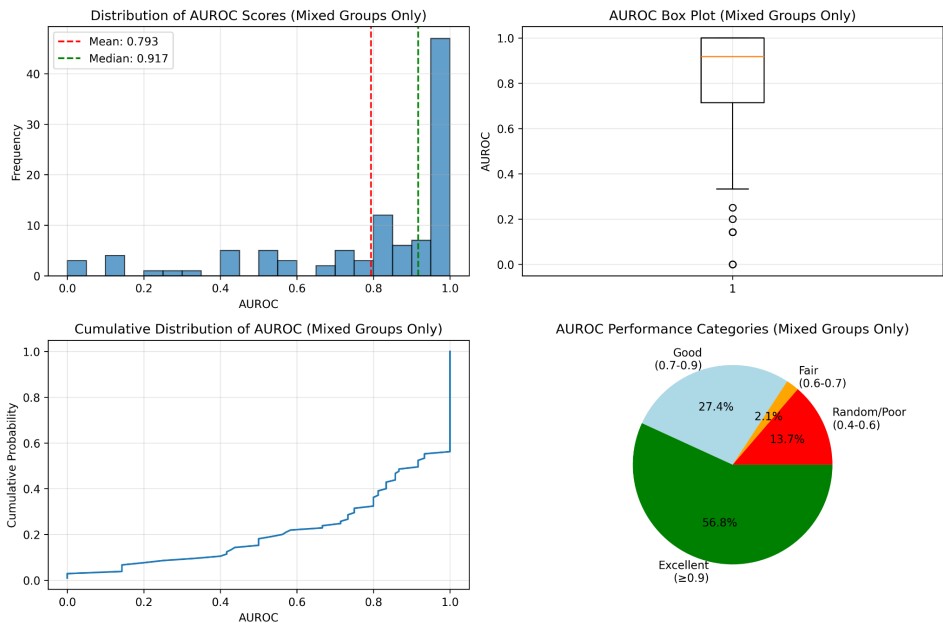

Figure 7: Reward model qualification ability on mixed groups: (a) distribution of AUROC scores, (b) AUROC box plot, (c) cumulative distribution of AUROC, and (d) AUROC performance categories.
s

## B.2  QUALITATIVE ANALYSIS OF RULE-BASED VERIFIERS

Table 10 highlights representative behaviors of rule-based and model-based verifiers. `math.py` is overly strict, failing on minor formatting variations such as boxing or punctuation (Rows 1–2), while `math_verify.py` improves recall through normalization. The `Math-Verify` library handles simple surface mismatches but struggles with structural differences like disjoint ranges or multiple valid tuples (Rows 4–5). In contrast, o3 is the most permissive: it credits partially correct sets (Row 3) and parametric families with renamed symbols (Row 6), which increases coverage but risks over-crediting. These cases illustrate the precision–recall trade-off: rule-based verifiers enforce exact symbolic correctness but miss semantically equivalent or partially correct answers, whereas model judges offer flexibility at the cost of reliability. This motivates our hybrid design: HERO anchors dense reward signals to rule-based correctness, ensuring robustness to format variance, while leveraging model- or RM-derived scores to provide graded feedback on harder cases involving subsets, orderings, or parametric equivalence.

## C  LIMITATIONS AND FUTURE WORK

While HERO demonstrates clear advantages over RM-only and verifier-only training, several limitations remain. First, the method depends on the availability and reliability of rule-based verifiers: when these are brittle or domain-mismatched, the partitioning into correctness groups may be biased, weakening the benefits of stratified normalization. More broadly, our method is explicitly designed for settings where a rule-based signal and a dense reward-model signal can be combined; when such a verifier is unavailable or highly unreliable, the current HERO formulation is not directly applicable. Second, because the reward model is trained primarily on outcome-based, verifiable data, it can become miscalibrated on harder, non-verifiable formats, and although our framework constrains its scores, residual bias or spurious correlations may still be exploited. Third, HERO introduces sensitivity to hyperparameters such as $(\alpha, \beta)$ and the weighting slope $k$, and increases training overhead due to concurrent verifier and RM calls. Finally, evaluation on non-verifiable tasks often relies on LLM-as-judge protocols, which introduce prompt sensitivity and annotation noise. Future work will focus on improving verifier coverage with hybrid symbolic–learned approaches, incorporating process-level supervision to capture reasoning quality beyond final answers, and developing adap-

| Ground truth | Model Prediction | math.py | math_verify.py(verl) | Math_verify library | o3 |
|---|---|---|---|---|---|
| $f(x) = 2x$ | `\boxed {f(x) = 2x}` | ✗ | ✓ | ✓ | ✓ |
| $(6, 3), (9, 3), (9, 5), (54, 5)$ | `\boxed {(6,3)}, \boxed {(9,3)}, \boxed {(9,5)}, \boxed {(54,5)}` | ✗ | ✓ | ✓ | ✓ |
| $(0, 1, 1), (0, -1, -1), (1, 0, 1),$ $(-1, 0, -1), (1, 1, 0), (-1, -1, 0),$ $\left(\frac{1}{\sqrt 3}, \frac{1}{\sqrt 3}, \frac{1}{\sqrt 3}\right),$ $\left(-\frac{1}{\sqrt 3}, -\frac{1}{\sqrt 3}, -\frac{1}{\sqrt 3}\right), \dots$ | `Final Answer: \boxed {(1,1,0)}, \boxed {(-1,-1,0)}, \boxed {(1/\sqrt {3},1/\sqrt {3},1/\sqrt {3})}, \boxed {(-1/\sqrt {3},-1/\sqrt {3},-1/\sqrt {3})},` | ✗ | ✗ | ✗ | ✓ |
| $10, 11, 12, 13, 14, -2, -1, 0, 1, 2$ | `Final Answer: \boxed {-2,-1,0,1,2} and \boxed {10,11,12,13,14}` | ✗ | ✓ | ✗ | ✓ |
| $(1, 7, 103, 105), (3, 5, 101, 107)$ | `Final Answer: Two possible lists are \boxed {(3,5,101,107)} and \boxed {(1,7,103,105)}` | ✗ | ✓ | ✗ | ✓ |
| $f(x) = ax + b$ (where b is an arbitrary integer, and a is an arbitrary positive integer with mho(a)=0) | `Final Answer: \boxed {f(n)=cn+d}, where c has no prime factors > 10ˆ{100} and d is any integer` | ✗ | ✓ | ✗ | ✓ |

Table 10: Examples demonstrating agreement between different math verification tools.

tive range and weighting schemes that calibrate dense signals online. These directions can further strengthen the stability and generality of hybrid reward frameworks for reasoning.

Mathematical reasoning benchmarks are a natural first testbed for HERO, as they provide mature, high-precision verifiable rewards (e.g., programmatic checks for final answers) and are widely used in recent RLVR systems such as DeepSeek-R1. For this reason, we intentionally restrict our empirical study to math reasoning, where HERO's assumptions are well satisfied and comparisons to existing verifier-based RL methods are most meaningful.In contrast, many non-mathematical, open-ended, or multimodal tasks currently lack clear, high-precision rule-based verifiers, making them less aligned with our present problem setting. We therefore do not claim that HERO, as instantiated in this paper, already solves open-ended generation; instead, these tasks remain outside the scope of our experiments. Nonetheless, HERO itself is agnostic to the specific form of $r_{\text{rule}}$: in principle, any structured checker (such as code-execution tests, safety or formatting constraints, or even rubric-guided LLM-as-judge signals) could play this role. Systematically extending HERO to such "soft" verifiers for open-ended and multimodal reasoning, and studying the resulting trade-offs between stability, coverage, and bias, is an important direction for future work.

# D THE USE OF LARGE LANGUAGE MODELS(LLM)

In our project, we use LLM for writing polishing.

