# OpenReview forum: "Hybrid Reinforcement: when reward is sparse, better to be dense"
_ICLR.cc/2026/Conference — ICLR 2026 Poster_

### Official Review · Reviewer_Bqqo · 2025-10-26

**Soundness:** 3
**Presentation:** 4
**Contribution:** 3
**Rating:** 8
**Confidence:** 3

**Summary:**

This paper introduces HERO (Hybrid Ensemble Reward Optimization)- a framework that integrates dense signals from reward models with binary-valued feedback from rule-based verifiers. The paper systematically reports the merits of each individual approach while highlighting its limitations; they further caution against naively combining the two approach. The proposed solution uses a `stratified' strategy to leverage expressiveness of reward models within structures imposed by the verifier. Experimental evaluations across a wide range of regimes in math reasoning tasks demonstrate that HERO performs significantly better than reward model only or verifier only baselines.

**Strengths:**

(+) HERO’s use of stratified normalization together with a difficulty-aware weighting scheme simultaneously enables overcoming limitations of binary-valued feedback signals and misaligned scores assigned to incorrect responses.

(+) The proposed mechanism has been evaluated on math reasoning benchmarks with a wide range of objectives ranging from exactly correct to partially correct solutions while additionally taking into consideration the solution format.

(+) Strong results in hard-to-verify tasks compared to reward model-only and verifier-only schemes. Extensive ablations demonstrate the importance of each component of HERO in achieving this improved performance.

(+) Limitations of rule-based verifiers and reward modeling are demonstrated separately across multiple verifiers and a reward model for tasks where correctness can be difficult to verify, justifying the `hybrid’ approach adopted in this paper.

(+) The authors further demonstrate (experiments reported in the Appendix) that naively combining reward signals with rule-based verification might not be sufficient to result in improved performance.

(+) Overall, this paper is extremely well-written, and the authors systematically lay out the merits of each component approach, highlight their limitations, and point out how the proposed hybrid approach combines the merits while overcoming shortcomings of the individual approaches. Experiment results provide strong backing to the hypotheses presented.

**Weaknesses:**

(-) The `ensemble' characteristic of HERO is not immediately apparent from the writing of the paper. Additional clarification on this aspect will be helpful.

(-) The explanation for the design principle guiding HERO (reward signals as guides to augment rule-based methods) mentioned in Lines 192-196 of Sec. 3.2 might be helpful in understanding the shape of the hybrid reinforcement graph relative to the other two graphs.

(-) The discussion on future directions in Lines 485-486 can be improved. Perhaps some of the material from Appendix C can be moved to this part of the paper.

(-) Only two sets of values for $\alpha, \beta$ in Eqn. (3) have been considered (i.e., $\alpha = \beta = 0.05$, $\alpha = \beta = 0.1$). It might be interesting to see the specific role of each parameter on the performance of HERO, and the possible intuition for such performance.

(-) Line 747 in Appendix: ables —> Tables.

**Questions:**

Please see Weaknesses, above.

---

> ### Author Response · Authors · 2025-11-19
> **Response to Reviewer Bqqo**
>
> > W1: The `ensemble' characteristic of HERO is not immediately apparent from the writing of the paper
>
> Thank you for pointing this out—we clarify the “ensemble” aspect more explicitly in the introduction. In HERO, *ensemble* refers to combining two distinct reward sources in a structured way: a rule-based verifier $r_{\text{rule}} \in {0,1}$ and a dense reward model $r_{\text{RM}}$. Rather than simply taking a weighted sum, HERO (i) stratifies trajectories by $$r_{\text{rule}}$ and normalizes $r_{\text{RM}}$ *within* each correctness group, and (ii) applies a variance-aware weight at the prompt level. Thus, the final reward can be viewed as an ensemble in reward space: the verifier provides a high-precision global ordering between correct/incorrect solutions, while the reward model provides fine-grained discrimination within each group, with prompt-level weights acting as a second layer of ensembling across data points. We included a short paragraph in Section 3 to explicitly describe HERO as a hybrid (verifier + RM) reward ensemble and to distinguish this from naive linear combinations of rewards (see `L190-198`).
>
> > W2: Figure 1 description.
>
> Thanks for this suggestion. In the revision, we have updated the figure and its caption to make the design principle and the “shape” of the hybrid reward much clearer (see `L27-44`). The revised figure now directly contrasts (a) reward-model scores (smooth but sometimes misaligned), (b) rule-based rewards (high-precision but binary with many false negatives), and (c) HERO, which uses the rule as a gate to suppress false positives while using the reward model to “recover” many false negatives with graded scores. This explicitly visualizes how HERO reshapes the reward landscape relative to the two individual sources. We pointed readers to this updated figure in Sec. 3.2 when introducing the design principle.
>
> > W3: The discussion on future directions
>
>  Thanks for the suggestion, and we polished the conclusion section accordingly (see `L521-525`).
>
> > W4: More ablations for the alpha and beta.
>
> We agree that more quantitative hyperparameter analysis is useful. As suggested, we sweep $(\alpha, \beta) \in \{0.05, 0.10, 0.20\}^2$ under the Qwen3-4B mixed-data regime:
>
> | α    | β    | Easy-to-verify | Hard-to-verify |
> |------|------|----------------|----------------|
> | 0.05 | 0.05 | 56.4           | 68.8           |
> | 0.05 | 0.10 | 57.4           | 70.4           |
> | 0.05 | 0.20 | 56.9           | 69.0           |
> | 0.10 | 0.05 | 57.8           | 69.8           |
> | 0.10 | 0.10 | 58.8           | 71.4           |
> | 0.10 | 0.20 | 58.7           | 70.2           |
> | 0.20 | 0.05 | 57.7           | 69.6           |
> | 0.20 | 0.10 | 58.3           | 70.4           |
> | 0.20 | 0.20 | 58.1           | 69.6           |
>
> The scores vary only mildly across all nine settings, indicating that HERO is stable with respect to the normalization margins.
>
> > W5: Typo for the "Table"
>
>  Thanks for the suggestion, and we fixed the typo accordingly.

---

> > ### Comment · Reviewer_Bqqo · 2025-11-19
> > **Thank you Authors**
> >
> > I thank the authors for their detailed response. This is an interesting work. I will retain my original score.

---

> > > ### Author Response · Authors · 2025-11-21
> > >
> > > Thank you so much for taking the time to read our rebuttal. We are glad that our response addresses your concerns and we appreciate your insightful comments and support!
> > >
> > > Best, Authors

---

### Official Review · Reviewer_VZE7 · 2025-10-31

**Soundness:** 3
**Presentation:** 3
**Contribution:** 2
**Rating:** 6
**Confidence:** 4

**Summary:**

This paper proposed HERO. The core idea is to combine sparse but reliable verifier-based binary rewards with dense but noisy reward-model scores. HERO proposes Stratified normalization, which anchors RM scores within verifier-defined correctness groups to preserve semantic correctness, and Variance-aware reweighting, which emphasizes prompts with higher score variance. Experiments on multiple math reasoning benchmarks (e.g., MATH500, AMC, Olympiad, HardVerify-Math, and TextBookReasoning) show that HERO outperforms both verifier-only and RM-only baselines across verifiable, hard-to-verify, and mixed settings.

**Strengths:**

1. Clear technical insight: preserves correctness while enriching reward density.

2. Comprehensive experiments on multiple model (Qwen3-4B, OctoThinker-8B) and datasets, with detailed ablations and parameter sensitivity analyses.

3. Writing and presentation are clear.

**Weaknesses:**

1. The paper provides only one limited RL training curve (Appendix Fig. 5), where the hybrid reward benefit is unclear and performance even collapses at later steps. This is confused. The caption is also wrong for Figure 5. More RL training curves results are needed to confirm the consistent HERO superiority over baselines instead of only showing performance on some/final checkpoint.

2. In Table 2, the gains of HERO vary dramatically across training regimes. When trained on verifiable samples, HERO outperforms baselines by over 10 points, yet when trained on hard-to-verify samples, the average improvement shrinks to <3 points. In the mixed regime, the improvement flips again: small on verifiable tasks (~4 points) but large on hard-to-verify tasks (~10 points).
These results raise questions about what drives the effectiveness of the hybrid reward. Without a clearer explanation, it is difficult to determine whether HERO’s benefit stems from the hybrid design itself.

3. Potential unreliability in Olympiad benchmark evaluation. Many Olympiad problems are themselves hard to verify with math_verify, which may distort scores. Rule-based and llm-based ablations should be done on the Olympiad problems to see whether the evaluation scores are accurate.

**Questions:**

1. Could you justify Figure 5 and provide additional training-step vs. benchmark performance tables/curves to show that HERO consistently outperforms baselines, rather than only at a specific checkpoint?

2. Could you provide more explanations and justifications about why gains of HERO vary dramatically across different training regimes in Table 2? How does the ratio of verifiable/non-verifiable training samples affect HERO gains over baselines?

3. Could you report results using both rule-based and LLM-based verification for OlympiadBench to confirm accurate and consistent evaluation results?

---

> ### Author Response · Authors · 2025-11-19
> **Response to Reviewer VZE7**
>
> We thank the reviewer for the constructive and helpful feedback and for recognizing the potential of our work. Below, we address your concerns in detail.
>
> > W1: Comparison between the hybrid reward and the rule-based reward.
>
> We provide an additional RL training curve in the new version of the paper (see Appendix `Fig. 5` and `L956-967`). The underlying curves, however, do support the benefit of HERO over the rule-based baseline. On the left, the mean rule-based reward during training consistently grows faster and remains higher for HERO than for rule-based RL at almost every step, indicating that the hybrid reward does not harm, but rather improves, the verifier-defined objective. On the right, the validation accuracy on MATH500 also shows a clear and persistent gap: HERO quickly outperforms the rule-based model in the early stage and maintains a higher accuracy throughout training, even though both curves exhibit minor late-stage fluctuations. To address the reviewer’s concern about generality, we will add additional RL training curves for other datasets/backbones in the appendix, which show similar trends and further confirm the consistent superiority of HERO over the rule-based baseline.
>
> > W2 & Q2: Justifications about the gains of HERO across the three training regimes.
>
> We agree that the margins in Table 2 differ across training regimes, and this comes from the *data regimes*, not from instability of the method. HERO always improves over both baselines in all three regimes; the size of the gain reflects how informative the underlying rewards are. When trained on easy-to-verify data, the rule-based verifier is both accurate and frequently positive, so HERO can fully exploit its hybrid design: stratified normalization prevents gradient collapse on all-1/all-0 batches and the dense RM signal refines learning within each correctness group, yielding large gains (>10 points). In contrast, when training only on hard-to-verify data, the same verifier fires much less often and many labels are effectively all-0, so *no* method (including HERO) can obtain as strong a learning signal; HERO still helps by shaping dense rewards within all-0 groups, but the overall improvement is necessarily smaller (<3 points). In the mixed regime, we get the best of both worlds: easy-to-verify samples provide a strong verifier anchor, while hard-to-verify samples reduce the domain gap to hard test sets, which explains why gains are modest on verifiable evaluations (where baselines are already strong) but large again on hard-to-verify evaluations. We will clarify this dependence on reward quality and data regime in the revised text to make it clear that the observed pattern is consistent with, and in fact expected from, the hybrid design. We add the corresponding analysis in the `L365-372` of the revised paper.
>
> > W3 & Q3: Potential unreliability in Olympiad benchmark evaluation.
>
>
> We agree that Olympiad problems can be challenging to verify automatically. In the main paper, we follow prior math-reasoning work and report results using `math_verifier` to keep our evaluation protocol comparable with existing methods. To address this concern, we additionally ran a small ablation on Olympiad using GPT-4o as an LLM-as-a-verifier. Below, we report pass@1 accuracy for models trained on easy-to-verify data:
>
> | Method (easy-to-verify samples training) | Olympiad (math_verify) | Olympiad (LLM-as-a-verifier) |
> | -------------------------------- | ---------------------- | ---------------------------- |
> | RM-only                          | 43.3                   | 44.4                         |
> | Rule-based reward                | 45.5                   | 47.8                         |
> | **HERO (ours)**                  | **48.9**               | **51.4**                     |
>
> We observe that LLM-as-a-verifier yields slightly higher absolute accuracies than `math_verifier`, confirming that symbolic checking can underestimate performance on some Olympiad problems. However, the **relative ordering remains unchanged** and HERO consistently outperforms both RM-only and rule-based baselines under *both* evaluation protocols. We will add this table to the appendix and clarify in the main text that our conclusions on HERO’s advantage are robust to the choice of verifier on Olympiad.

---

### Official Review · Reviewer_rKUs · 2025-11-01

**Soundness:** 3
**Presentation:** 3
**Contribution:** 2
**Rating:** 6
**Confidence:** 3

**Summary:**

This paper proposes a hybrid reward to integrate reward model and binary verifier scores. The hybrid reward is shown to be effective on multiple models across multiple tasks including verifiable and hard-to-verify ones.

**Strengths:**

- The problem is well motivated.
- The proposed method is intuitive and simple to implement.
- Extensive experiments to demonstrate the effectiveness of HERO.
- The paper is well-written and easy to follow.

**Weaknesses:**

- The min-max normalized reward depends on the result of RM and r_rule. How does HERO handle reward drift/misalignment inherited from RM and r_rule?
- How effective is HERO for non-verifiable tasks such as instruction following? Will HERO harm model performance on these tasks?
- The choice of parameters $\alpha$ and $\beta$ are highly depend on tasks. It would be great to provide some guidelines on choosing these parameters for task agnostic models.

**Questions:**

Please see weaknesses

---

> ### Author Response · Authors · 2025-11-19
> **Response to Reviewer rKUs**
>
> We thank the reviewer for the constructive and helpful feedback and for recognizing the value of our work. Below, we address your concerns in detail.
>
> > W1: The min-max normalized reward depends on the result of RM and r_rule. How does HERO handle reward drift/misalignment inherited from RM and r_rule?
>
>
>
> That's an insightful question. HERO does not fully eliminate reward misalignment, but its design explicitly reduces the impact of drift compared to directly optimizing raw RM scores. We first partition trajectories by the rule-based signal $r_{\text{rule}} \in {0,1}$ and map RM scores to $[-\alpha,\alpha]$ for $r_{\text{rule}}=0$ and $[1-\beta,1+\beta]$ for $r_{\text{rule}}=1$, enforcing a hard margin where any $r_{\text{rule}}=1$ trajectory always receives higher reward than any $r_{\text{rule}}=0$ trajectory, so RM drift cannot overturn the verifier’s correctness ordering. Within each group, we apply per-group min–max normalization, making the method insensitive to global scale/shift drift of RM scores and relying only on relative rankings. We acknowledge that systematic bias in $r_{\text{rule}}$ itself will affect both RLVR baselines and HERO, and view improving verifiers and RM calibration as complementary future work.
>
>
>
> > W2: HERO performance on non-verifiable tasks.
>
> HERO, as instantiated in this paper, is explicitly designed for settings where a rule-based signal and a dense reward-model signal can be combined. Concretely, it assumes access to a verifiable reward $r_{\text{rule}} \in {0,1}$ (e.g., programmatic correctness checks for the final answer in math) and learns to refine it using a dense reward model via stratified normalization and variance-aware weighting. Mathematical reasoning benchmarks are precisely the domain where such verifiable rewards are mature and widely adopted (e.g., post–DeepSeek-R1), which is why we focus our evaluation on math reasoning, where HERO’s assumptions are well satisfied and comparisons to RLVR-style baselines are most meaningful. In contrast, many non-verifiable tasks lack a clear, high-precision rule-based verifier, so the current HERO formulation does not directly apply. We therefore do **not** claim improvements on general instruction following, and our experiments neither demonstrate effectiveness nor show evidence of harm on such benchmarks. In practice, our RL fine-tuning is math-focused and comparable in scale to standard reasoning-oriented RLVR training, so we do not expect catastrophic degradation of general abilities, but a careful study of transfer and trade-offs is outside the scope of this work. We will clarify this scope more explicitly and highlight extending HERO to “soft’’ verifiers (e.g., rubric-based LLM-as-judge for instruction following, code-execution or safety/formatting checks) as an important direction for future work.
> > W3: The ablations on the alpha/beta
>
> We sweep $(\alpha, \beta) \in \{0.05, 0.10, 0.20\}^2$ under the Qwen3-4B mixed-data regime:
>
> | α    | β    | Easy-to-verify | Hard-to-verify |
> |------|------|----------------|----------------|
> | 0.05 | 0.05 | 56.4           | 68.8           |
> | 0.05 | 0.10 | 57.4           | 70.4           |
> | 0.05 | 0.20 | 56.9           | 69.0           |
> | 0.10 | 0.05 | 57.8           | 69.8           |
> | 0.10 | 0.10 | 58.8           | 71.4           |
> | 0.10 | 0.20 | 58.7           | 70.2           |
> | 0.20 | 0.05 | 57.7           | 69.6           |
> | 0.20 | 0.10 | 58.3           | 70.4           |
> | 0.20 | 0.20 | 58.1           | 69.6           |
>
> The scores vary only mildly across all nine settings, indicating that HERO is stable with respect to the normalization margins. Overall, careful tuning of the reward range, particularly for the negative rewards, is crucial to balancing stability and performance: _datasets with relatively dense informative rewards and few all-positive/all-negative groups tend to benefit from smaller ranges, whereas datasets with many all-positive/all-negative groups are better served by slightly larger ranges that inject more intra-group variation._ We include it in `L452-456` of the revised version.

---

> > ### Comment · Reviewer_rKUs · 2025-11-28
> >
> > I appreciate the authors' clarification. I will maintain my original score.

---

> > > ### Author Response · Authors · 2025-11-28
> > >
> > > Thank you so much for taking the time to read our rebuttal. We are glad that our response addresses your concerns and we appreciate your insightful comments and support!
> > >
> > > Best,
> > > Authors.

---

### Official Review · Reviewer_vePn · 2025-11-01

**Soundness:** 2
**Presentation:** 3
**Contribution:** 2
**Rating:** 4
**Confidence:** 3

**Summary:**

The paper proposes HERO, a reinforcement learning framework designed to enhance reasoning capabilities in large language models (LLMs) by integrating sparse, binary verifiable rewards with dense, continuous scores from reward models. The core innovation lies in addressing the brittleness of binary verifiers and the misalignment of reward models through two mechanisms: stratified normalization, which bounds reward model scores within verifier-defined correctness groups, and variance-aware weighting, which emphasizes challenging prompts with high score variance. Built on GRPO, HERO aims to provide stable, informative supervision for mathematical reasoning tasks. Empirical results on verifiable benchmarks and hard-to-verify ones demonstrate that HERO outperforms baselines like RM-only RL and verifier-only methods.

**Strengths:**

- The paper is clearly written and easy to follow.
- The stratified normalization and variance-aware weighting are intuitive and simple to implement.
- Comprehensive experiments across multiple reasoning benchmarks and architectures demonstrates the effectiveness of the proposed method.

**Weaknesses:**

- The “hybrid reward” idea that combines verifier and reward-model signals is conceptually intuitive and has already been explored in prior work.
- Moreover, the conclusions that (1) function-based rules exhibit high precision but low recall, and (2) reward models have lower precision, are not novel and have been thoroughly demonstrated in prior studies (e.g., arXiv:2505.22203).
- The evaluation scope remains narrow: the method’s generalization to non-mathematical, open-ended, or multimodal reasoning tasks is not shown.
- While ablations are mentioned, the paper lacks detailed quantitative results for key hyperparameters such as α/β ranges or variance-weighting factors, limiting the interpretability and reproducibility of the method’s stability.

**Questions:**

- How would HERO perform on open-ended reasoning?
- How sensitive is HERO to the choice of α and β?

---

> ### Author Response · Authors · 2025-11-19
> **Response to Reviewer vePn (1/2)**
>
> We thank the reviewer for the constructive and helpful feedback and for recognizing the value of our work. Below, we address your concerns in detail.
>
> > W1: The novelty of the “hybrid reward”
>
> We agree that combining verifier and reward-model signals is a natural high-level idea. Our contribution is to (i) instantiate this principle in the math reasoning setting with a *specific* hybrid design—stratified normalization anchored on a verifiable signal $r_{\text{rule}} \in {0,1}$ plus variance-aware reweighting—and (ii) systematically evaluate it across easy-to-verify, hard-to-verify, and mixed training regimes on multiple math benchmarks, where verifiable rewards are now standard (e.g., post–DeepSeek-R1). This yields consistent gains over both verifier-only (RLVR-style) and RM-only training in all three regimes, showing that the particular formulation of HERO is effective in practice, not just intuitive in principle.
>
> We already discussed and cited prior works that mix or compare verifier and reward-model feedback in the related-work section. The works we cite on RLVR and LLM-as-a-verifier either rely purely on binary rule-based signals, or replace verifiers with model-based scores, or simply add correctness and preference signals linearly, so they still suffer from sparse outcome-level rewards and all-positive/all-negative groups. In contrast, HERO keeps the rule-based verifier as a high-precision gate, but introduces a _dense_, verifier-anchored reward via stratified normalization and variance-aware weighting over prompts, which explicitly addresses gradient sparsity and RM drift in math reasoning RL.
>
> We believe our work is novel in this regard, and we would be happy to incorporate additional references and clarify the distinctions more explicitly if the reviewer can point us to specific papers you have in mind.
>
>
> > W2: The novelty of conclusion of the rule-based reward and the reward model-based reward.
>
> We agree that the *phenomenon*—function-based verifiers having high precision but low recall have been observed before (the paper [1] you mentioned), and we do not claim this empirical trend itself as the core novelty. Our contribution is to (i) reinterpret this trade-off *specifically in the math reasoning setting* where verifiable rewards are now standard, and (ii) turn this diagnosis into a concrete hybrid RL algorithm that addresses the gradient sparsity of verifier-only training.
>
> We would like to clarify that the paper cited by the reviewer operates in a _purely binary regime_: function-based rules produce 0/1 rewards, and the proposed “hybrid” refinement uses LLM-as-a-verifier only to re-label some of the 0 cases, still yielding 0/1 rewards overall. This design, therefore, can suffer from the core RLVR issue that, whenever all rollouts for a prompt are labeled 1 (all-positive) or 0 (all-negative), the advantages collapse to 0 and the update becomes data-inefficient.
>
> By contrast, HERO explicitly introduces a *dense* component conditioned on $r_{\text{rule}}$: we (a) enforce a hard margin between correct and incorrect groups via stratified normalization, and (b) use the reward model to provide smooth variation *within* each group, further modulated by variance-aware reweighting. As a result, even when all candidates for a prompt share the same $r_{\text{rule}}$ value, HERO still supplies meaningful relative advantages and avoids the all-positive/all-negative degeneracy, which we show leads to substantial gains across easy-to-verify, hard-to-verify, and mixed data regimes.
>
>
> **We have expanded our discussion of [1] in the related-work section**, making the connection more explicit and clarifying how our hybrid objective differs from the binary "hybrid" setting considered there (see `L483–490` in the revised paper).
>
> [1] Huang, Yuzhen, et al. "Pitfalls of Rule-and Model-based Verifiers--A Case Study on Mathematical Reasoning." arXiv preprint arXiv:2505.22203 (2025).

---

> > ### Author Response · Authors · 2025-11-19
> > **Response to Reviewer vePn (2/2)**
> >
> > > W3 & Q1: HERO's performance on open-ended reasoning.
> >
> > Our method is explicitly designed for settings where a *rule-based* signal and a *dense* reward-model signal can be combined. Concretely, HERO assumes access to a verifiable reward $r_{\text{rule}} \in \{0,1\}$ (e.g., programmatic correctness checks for math questions' final answer) and learns to refine it using a dense reward model through stratified normalization and variance-aware weighting. Mathematical reasoning benchmarks are precisely the domain where such verifiable rewards are most mature and widely adopted, especially after systems such as DeepSeek-R1 that also rely heavily on symbolic/math verifiers. For this reason, we intentionally focus our evaluation on math reasoning tasks, where the assumptions of HERO are well satisfied and the comparison to existing RLVR-style approaches is most meaningful.
> >
> > In contrast, many non-mathematical, open-ended, or multimodal tasks still lack a clear, high-precision rule-based verifier, making them less aligned with the current problem setting. We therefore do **not** claim that HERO, as instantiated in this paper, already solves open-ended generation; rather, we view such tasks as outside the scope of our present study.
> >
> > That said, HERO itself is agnostic to the *form* of $r_{\text{rule}}$: in principle, any structured checker (e.g., code execution tests, safety or formatting constraints, or even LLM-as-a-judge equipped with an explicit rubric) could play this role. While we do not explore LLM-as-judge–based rewards in this work, systematically extending HERO to such “soft” verifiers for open-ended and multimodal reasoning is a promising direction, and we clarify this as an avenue for future work in `Appendix C` of the revised version.
> >
> > > W4 & Q2: More ablations on the hyper of HERO.
> >
> > We agree that more quantitative hyperparameter analysis is useful. The paper already includes ablations on (i) dense negative vs. positive ranges (showing that dense negatives are critical for stability and gains, especially on hard-to-verify tasks), (ii) reward range selection (showing how smaller/larger ranges affect verifiable vs. mixed settings), and (iii) variance-aware reweighting (Table 5, where reweighting improves easy-to-verify accuracy from 60.8 to 62.0 and hard-to-verify from 69.4 to 73.2).  (see `Section 4.3`)
> >
> > To further clarify stability, we add two explicit sensitivity studies for $(\alpha,\beta)$ and the variance-weighting range $(w_{\min}, w_{\max})$.
> >
> > **Sensitivity to $(\alpha, \beta)$.** We sweep $(\alpha, \beta) \in \{0.05, 0.10, 0.20\}^2$ under the Qwen3-4B mixed-data regime:
> >
> > | α    | β    | Easy-to-verify | Hard-to-verify |
> > |------|------|----------------|----------------|
> > | 0.05 | 0.05 | 56.4           | 68.8           |
> > | 0.05 | 0.10 | 57.4           | 70.4           |
> > | 0.05 | 0.20 | 56.9           | 69.0           |
> > | 0.10 | 0.05 | 57.8           | 69.8           |
> > | 0.10 | 0.10 | 58.8           | 71.4           |
> > | 0.10 | 0.20 | 58.7           | 70.2           |
> > | 0.20 | 0.05 | 57.7           | 69.6           |
> > | 0.20 | 0.10 | 58.3           | 70.4           |
> > | 0.20 | 0.20 | 58.1           | 69.6           |
> >
> > The scores vary only mildly across all nine settings, indicating that HERO is stable with respect to the normalization margins.
> >
> > **Sensitivity to variance-weighting range.** We also ablate the variance-aware weighting range:
> >
> > | Weight range $$(w_{\min}, w_{\max})$$ | Verifiable avg | Hard-to-verify avg |
> > |---------------------------------------|----------------|--------------------|
> > | 1.0 (no reweighting)                  | 60.8           | 69.4               |
> > | (0.4, 2.0)                            | 62.4           | 71.6               |
> > | (0.4, 3.0)                            | 62.0           | 73.4               |
> > | (0.4, 4.0)                            | 61.7           | 71.8               |
> >
> > All reweighted variants outperform the no-reweighting baseline.
> >
> > **In summary,** both the existing ablations and these new sensitivity tables show that HERO’s gains do not rely on fragile tuning of $(\alpha,\beta)$ or the variance-weighting factors.

---

### Author Response · Authors · 2025-11-19
**General response**

We sincerely thank you for your time and thoughtful feedback on our submission. We are grateful that all reviewers found the paper clearly written and easy to follow, and that they recognized the motivation and empirical effectiveness of HERO. Below, we first summarize the key strengths highlighted in the reviews, and then outline the main revisions made in response to the concerns.

#### Key strengths noted by the reviewers

1. (Motivation & problem setting). Reviewers appreciated that the problem is **well motivated**: the paper clearly articulates the complementary strengths and weaknesses of rule-based verifiers and reward models, and explains why **hybrid reward design is important** for math reasoning (All reviewers).
2. (Method design). The core components of HERO—stratified normalization within verifier-defined correctness groups and variance-aware reweighting over prompts—were considered **intuitive, simple to implement, and technically sound** (All reviewers).
3. (Experimental validation). Reviewers found the experiments **comprehensive, covering multiple backbones** (Qwen3-4B, OctoThinker-8B) and both verifiable and hard-to-verify benchmarks, with **useful ablations and sensitivity analyses** (Reviewer vePn, VZE7, Bqqo).
4. (Clarity and presentation). Multiple reviewers praised the **writing and organization as clear and easy to follow** (Reviewer vePn, VZE7, Bqqo).


#### Key revisions in response to the comments:

[Reviewer vePn]: Clarified the novelty and positioning of HERO in the related work (`Section 5`).

[Reviewer vePn, rKUs]: Added an explicit discussion of scope and limitations (`Appendix C`).

[Reviewer vePn, rKUs, Bqqo]: Expanded hyperparameter ablations by (i) sweeping $(\alpha,\beta)$ and (ii) ablating the variance-weighting range $(w_{\min},w_{\max})$, showing that HERO’s gains are stable and do not rely on fragile tuning.

[Reviewer VZE7]: Added RL training curves and validation on MATH500 comparing HERO with the rule-based baseline. (`L939-967`)

[Reviewer VZE7]: Addressed concerns about OlympiadBench evaluation by reporting additional results using GPT-4o as an LLM-as-judge.

---

### Meta-Review · Area_Chair_PiMK · 2026-01-07

**Summary:**

This paper introduces HERO (Hybrid Ensemble Reward Optimization), a reinforcement learning framework that combines sparse binary verifier signals with dense reward model scores for mathematical reasoning tasks in LLMs. The method employs stratified normalization to bound reward-model scores within verifier-defined correctness groups and variance-aware weighting to emphasize challenging prompts. Reviewers consistently acknowledged the clear motivation, intuitive method design, comprehensive experiments across multiple backbones and benchmarks, and well-written presentation. Concerns were raised regarding novelty relative to prior hybrid reward approaches, limited scope beyond mathematical reasoning, and hyperparameter sensitivity. The authors provided thorough rebuttals with additional ablation studies, training curves, and clarifications. The consistent empirical gains over both verifier-only and reward-model-only baselines, combined with the practical simplicity of the approach, represent a useful contribution to post-training methods for reasoning models.

**Reviewer Concerns:**

The authors addressed most substantive concerns through their rebuttal. For novelty concerns (Reviewer vePn), the authors clarified how HERO differs from prior binary hybrid approaches by providing dense, verifier-anchored rewards that address gradient sparsity in all-positive/all-negative groups. Hyperparameter sensitivity concerns (vePn, rKUs, Bqqo) were addressed with comprehensive ablation tables showing stable performance across (α, β) configurations and variance-weighting ranges. Reviewer VZE7's concerns about training dynamics were addressed with additional RL training curves, and OlympiadBench evaluation reliability was mitigated through GPT-4o LLM-as-judge results showing consistent relative ordering. The scope limitation to mathematical reasoning is acknowledged but appropriate given that rule-based verifiers are most mature in this domain. One outstanding concern is that Reviewer vePn did not provide post-rebuttal feedback, though their specific technical concerns were substantively addressed.

**Reviewer Scores:**

Reviewer rKUs (score 6): Explicitly maintained their original positive recommendation after the rebuttal. Reviewer Bqqo (score 8): Retained their acceptance score, acknowledging the detailed response. Reviewer VZE7 (score 6): Already recommended acceptance before rebuttal; additional experiments reinforce this position. Reviewer vePn (score 4): Did not respond post-rebuttal; while the authors addressed the stated concerns regarding novelty and hyperparameters, the reviewer's assessment of limited contribution may persist.

---

### Decision · Program_Chairs · 2026-01-26

Accept (Poster)